# Efficient capture and storage of ammonia in robust aluminium-based metal-organic frameworks

Lixia Guo [1], Joseph Hurd [2], Meng He [1], Wanpeng Lu[1], Jiangnan Li [1], Danielle Crawshaw[1], Mengtian Fan[1], Sergei Sapchenko[1], Yinlin Chen[1], Xiangdi Zeng[1], Meredydd Kippax-Jones[1,3], Wenyuan Huang[1], Zhaodong Zhu[1], Pascal Manuel[4], Mark D. Frogley[3], Daniel Lee [2], Martin Schröder [1✉] & Sihai Yang [1✉]

The development of stable sorbent materials to deliver reversible adsorption of ammonia ($NH_3$) is a challenging task. Here, we report the efficient capture and storage of $NH_3$ in a series of robust microporous aluminium-based metal-organic framework materials, namely MIL-160, CAU-10-H, Al-fum, and MIL-53(Al). In particular, MIL-160 shows high uptakes of $NH_3$ of 4.8 and 12.8 mmol $g^{-1}$ at both low and high pressure (0.001 and 1.0 bar, respectively) at 298 K. The combination of in situ neutron powder diffraction, synchrotron infrared micro-spectroscopy and solid-state nuclear magnetic resonance spectroscopy reveals the preferred adsorption domains of $NH_3$ molecules in MIL-160, with H/D site-exchange between the host and guest and an unusual distortion of the local structure of $[AlO_6]$ moieties being observed. Dynamic breakthrough experiments confirm the excellent ability of MIL-160 to capture of $NH_3$ with a dynamic uptake of 4.2 mmol $g^{-1}$ at 1000 ppm. The combination of high porosity, pore aperture size and multiple binding sites promotes the significant binding affinity and capacity for $NH_3$, which makes it a promising candidate for practical applications.

[1] Department of Chemistry, University of Manchester, Manchester M13 9PL, UK. [2] Department of Chemical Engineering, University of Manchester, Manchester M13 9PL, UK. [3] Diamond Light Source, Harwell Science and Innovation Campus, Oxfordshire OX11 0DE, UK. [4] ISIS Facility, STFC Rutherford Appleton Laboratory, Chilton, Oxfordshire OX11 0QX, UK. ✉email: M.Schroder@manchester.ac.uk; Sihai.Yang@manchester.ac.uk

Ammonia ($NH_3$) is an essential feedstock for agriculture and industry and is currently being produced at a scale of approximately 180 million tonnes annually[1]. Moreover, the high density of $H_2$ (17.7 wt% gravimetrically and 123 kg m$^{-3}$ volumetrically) within $NH_3$ make the latter an attractive surrogate $H_2$ storage medium[2]. However, because of its toxic and corrosive nature, exposure to $NH_3$ is detrimental to the environment and health[3], and thus porous sorbents that are capable of removing trace $NH_3$ and exhibit high $NH_3$ uptakes are of great interest for air remediation and $NH_3$ storage[4,5]. The adsorbents must display high affinity to $NH_3$ to allow adsorption at low pressures and/or low concentrations[6]. Conventional materials such as activated carbons[7] and organic polymers[8], usually suffer from low adsorption affinity, poor stability and/or low uptakes for $NH_3$ adsorption. Therefore, it is an important but challenging task to design new materials with simultaneously high affinity, uptake and stability for adsorption of $NH_3$.

Metal-organic frameworks (MOFs) have emerged as excellent adsorbents for $NH_3$ that surpass the performance of conventional sorbents owing to their high porosity and tuneable structure[9]. MOFs with varying pore size and functional groups have been explored, and the optimisation of pore environment is key to improve the adsorption of $NH_3$. For example, the $\mu_2$-OH moieties in MFM-300(M) (M = Al, Fe, V, Cr, In, Sc)[2,10,11], and –COOH and $\mu_2$-OH groups in MFM-303(Al)[12] can act as the primary binding sites to promote the adsorption of $NH_3$. MOFs incorporating unsaturated metal sites can also exhibit strong adsorption of $NH_3$ at low pressure owing to the strong host−guest interactions, but they often show severe structural degradation upon desorption[13–16]. Al-based MOFs received much interest in adsorption of corrosive gases due to their high chemical stability and inexpensive synthesis at large scale. Nevertheless, their performance for adsorption of $NH_3$ has only been investigated to date in exceptional cases[10,12,17], and robust Al-MOFs showing high uptakes of $NH_3$ at both low and high pressures are yet to be developed.

Herein, we report the study of $NH_3$ adsorption in four Al-MOFs, namely, MIL-160[18], CAU-10-H[19], Al-fum[20], and MIL-53(Al)[21], incorporating distinct functional groups and structures. Specifically, the microporosity, abundant functional groups within the pores, and stability of MIL-160 make it promising for the study of capture and storage of $NH_3$, promoted by strong host−guest interactions and confinement effects. At 298 K, MIL-160

shows high uptakes of $NH_3$ of 4.8 and 12.8 mmol g$^{-1}$ at 0.001 and 1.0 bar, respectively. Dynamic breakthrough experiments confirm the excellent capability of MIL-160 for $NH_3$ capture at low concentration (1000 ppm) with a high dynamic uptake of 4.2 mmol g$^{-1}$, consistent with the high observed isosteric heats of adsorption (45-63 kJ mol$^{-1}$). The strong binding of $NH_3$ molecules to the $\mu_2$-OH groups and the heteroatom of the furan linker was directly visualised at crystallographic resolution via a combination of in situ neutron powder diffraction (NPD), synchrotron infrared micro-spectroscopy (SRIR) and solid-state nuclear magnetic resonance (ssNMR) spectroscopy. The host−guest interactions also impact the local structure of the MOF upon $NH_3$ binding, leading to distortions of [$AlO_6$] moieties, representing the first example of such an observation in a MOF studied by $^{27}$Al ssNMR. This work demonstrates the promising potential of robust Al-based MOFs for high and reversible adsorption of $NH_3$.

## Results and discussion

**Materials and characterisation.** The size, shape and chemical environment of the pores within a porous framework impact directly and control the adsorption of gas molecules. MIL-160[18], CAU-10-H[19], Al-fum[20] and MIL-53(Al)[21] were selected to investigate the effects of pore geometry, binding sites and the rigidity of framework on adsorption of $NH_3$ (Fig. 1). In each of these four materials, the Al(III) centre is bound by six O atoms from two hydroxyl and four carboxylate groups to give an [$AlO_6$] octahedral geometry. The frameworks in MIL-160 and CAU-10-H contain 4-fold helical chains comprised of corner-sharing [$AlO_4(OH)_2$] octahedral moieties linked through $cis$-$\mu_2$-OH bridges and bent linkers $H_2$fdc (2,5-furandicarboxylic acid) and $m$-$H_2$bdc (isophthalic acid), respectively. The "wine-rack" structures of MIL-160 and CAU-10-H give rise to square-shaped 1D channels of 5−6 Å diameter running along the $c$ axis (Fig. 1). By altering the linkers (from bent to linear), a distinct type of framework is formed for the isostructural Al-fum and MIL-53(Al) comprising of chains of $trans$-corner-sharing [$AlO_6$] octahedra linked with $H_2$fum (fumaric acid) and $p$-$H_2$bdc (terephthalic acid) ligands, respectively, to form 1D rhomb-shaped channels. MIL-53(Al) can reveal the impact of framework flexibility on $NH_3$ adsorption, while MIL-160, decorated with both $\mu_2$-OH groups and heteroatom oxygen centres in the pore, affords additional active sites. Importantly, the synthesis of MIL-160 is compatible with industrial requirements using water as the solvent and a biomass-derived organic linker. We hypothesised that the $\pi$-electrons of the furan rings and narrow micropores of MIL-160 would contribute to strong surface electrostatic interactions that would be beneficial for adsorption of $NH_3$.

The phase purity of all bulk materials was confirmed by powder X-ray diffraction (PXRD) (Supplementary Fig. 1). CAU-10-H shows a PXRD pattern that is nearly identical to that of MIL-160 due to the same $yfm$ topology[18,19]. The PXRD pattern of as-synthesised Al-fum shows a broad peak at ca. 20 degrees, which is consistent with reported work[20]. The thermal stability of these Al-MOFs was evaluated by thermogravimetric analysis (TGA) (Supplementary Fig. 2). In air, these Al-MOFs show thermal decomposition at 350 °C. The permanent microporosities of these Al-MOFs were evaluated by $N_2$ isotherms at 77 K, and all four MOFs exhibit a fully reversible type I adsorption profile (Supplementary Figs. 3−6a) with Brunauer−Emmett−Teller (BET) surface areas of 1000, 680, 1050, and 955 m$^2$ g$^{-1}$ for desolvated MIL-160, CAU-10-H, Al-fum and MIL-53(Al), respectively. These results are consistent with those previously reported for these materials[18–21]. The pore size distributions (PSD) were assessed according to the Horvath−Kawazoe cylinder model (Supplementary Figs. 3−6b). All

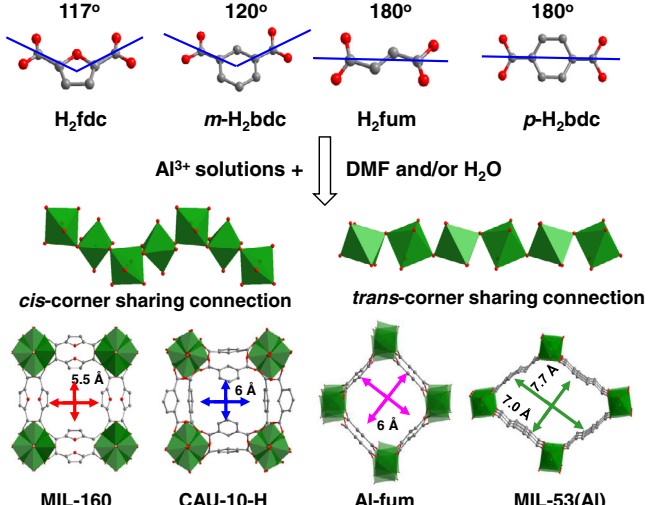

**Fig. 1 Schematic illustration of aluminium-MOFs.** Schematic illustration of selected linkers, the self-assembly processes through $cis$- and/or $trans$-$\mu_2$-OH connected [$AlO_6$] octahedral and the resulting MOFs.

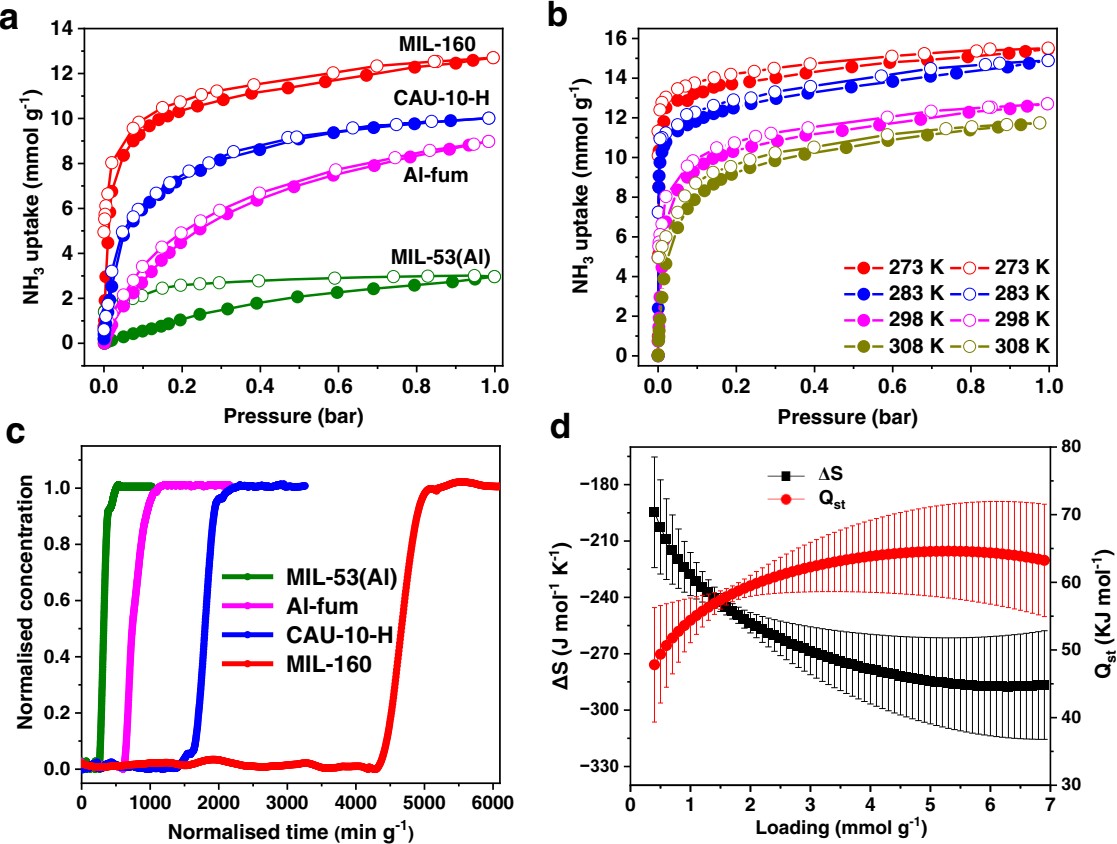

**Fig. 2 Adsorption, separation thermodynamics data. a** Adsorption and desorption isotherms for four Al-MOFs at 298 K (red: MIL-160; blue: CAU-10-H; magenta: Al-fum; olive: MIL-53(Al); solid: adsorption; open: desorption). **b** Adsorption and desorption isotherms for MIL-160 at 273−308 K (red: 273 K; blue: 283 K; magenta: 298 K; dark yellow: 308 K; solid: adsorption; open: desorption). **c** Dynamic breakthrough plots for $NH_3$ (1000 ppm diluted in He) with an inlet gas flow rate of 25 mL min$^{-1}$ through a fixed-bed packed with (olive) MIL-53(Al), (magenta) Al-fum, (blue) CAU-10-H and (red) MIL-160 samples at 298 K. **d** Plots for isosteric heats of adsorption ($Q_{st}$) and entropies of adsorption ($\Delta S$) (red: $Q_{st}$; black: $\Delta S$). The error bars were derived by least-squares linear fitting from four isotherms at different temperatures.

MOFs display narrow PSDs with main distribution centred at 5.5 Å (MIL-160), 6.0 Å (CAU-10-H), 6.0 Å (Al-fum) and 7.0/7.7 Å [MIL-53(Al)], consistent with the pore size determined from the crystal structure.

**Gas adsorption isotherms and breakthrough experiments.** Adsorption−desorption isotherms at 298 K and 0.001/1.0 bar show $NH_3$ uptake of 4.8/12.8, 1.4/10.0, 0.47/9.0, and 0.07/3.0 mmol g$^{-1}$ for MIL-160, CAU-10-H, Al-fum, and MIL-53(Al), respectively (Fig. 2a). The higher uptake of $NH_3$ in CAU-10-H compared with Al-fum and MIL-53(Al) suggests, not unexpectedly, that the surface area is not a direct indicator for $NH_3$ adsorption. Instead, the synergetic effect of pore geometry, rigidness of framework, and binding sites (e.g., $\mu_2$-OH) in the framework plays an important role in the total adsorption capacity[8]. With abundant π-electrons of the furan rings, high-density hydrogen-bonding nanotraps[22] and narrow micropores, MIL-160 exhibits the highest uptake of $NH_3$ among these four MOFs of 4.8/12.8 mmol g$^{-1}$ (at 298 K) and 6.9/15.5 mmol g$^{-1}$ (at 273 K) at 0.001/1.0 bar (Fig. 2b). The uptake of $NH_3$ at low pressures in MIL-160 exceeds those of best-behaving materials, such as MFM-300(Sc)[2] (2.0 mmol g$^{-1}$ at 0.001 bar at 273 K) and MFM-303(Al)[12] [6.0 and 8.3 mmol g$^{-1}$ at 0.002 bar and 273 K for MFM-303(Al) and MIL-160, respectively], indicating its excellent potential for capture of $NH_3$ at low concentrations.

The ability of MIL-160, CAU-10-H, Al-fum and MIL-53(Al) to capture $NH_3$ at 1000 ppm (diluted in He) was evaluated by dynamic breakthrough experiments at 298 K, and dynamic capacities were calculated to be 4.2, 1.3, 0.4 and 0.15 mmol g$^{-1}$, respectively (Fig. 2c and Supplementary Table 1). These values are consistent with the static, low-pressure capacities obtained from the isotherms at 298 K at 0.001 bar (4.8, 1.5, 0.5 and 0.17 mmol g$^{-1}$). The high dynamic uptake for MIL-160 suggests the presence of strong interactions between $NH_3$ and framework. The isosteric heats of adsorption ($Q_{st}$) of MIL-160 increases from 45 kJ mol$^{-1}$ to 63 kJ mol$^{-1}$ with increasing loading of $NH_3$ (Fig. 2d, Supplementary Fig. 7 and Supplementary Note 1), higher than that of MFM-303(Al) (61.5 kJ mol$^{-1}$)[12] and UiO-66-Cu$^{II}$ (25 − 55 kJ mol$^{-1}$)[16] (Supplementary Table 2), consistent with the presence of strong host−guest interactions and the observed high uptakes at low pressures. The $NH_3$-temperature programmed desorption (TPD) plot for MIL-160 shows that the adsorbed $NH_3$ molecules could be removed at around 200 °C (Supplementary Fig. 8), further confirming the strong host−guest interactions between $NH_3$ and MIL-160. In addition, a more negative entropy of adsorption ($\Delta S$) was observed in MIL-160 compared with other reported MOFs[10,12,16], suggesting a higher degree of order of adsorbed $NH_3$ molecules within the framework.

**Regeneration and stability test.** MIL-160 also shows a high packing density of $NH_3$ of 0.59 g cm$^{-3}$ at 273 K, comparable to that of liquid $NH_3$ (0.68 g cm$^{-3}$) at 240 K[16] and that of top-performing MOFs (Supplementary Table 3), suggesting MIL-160

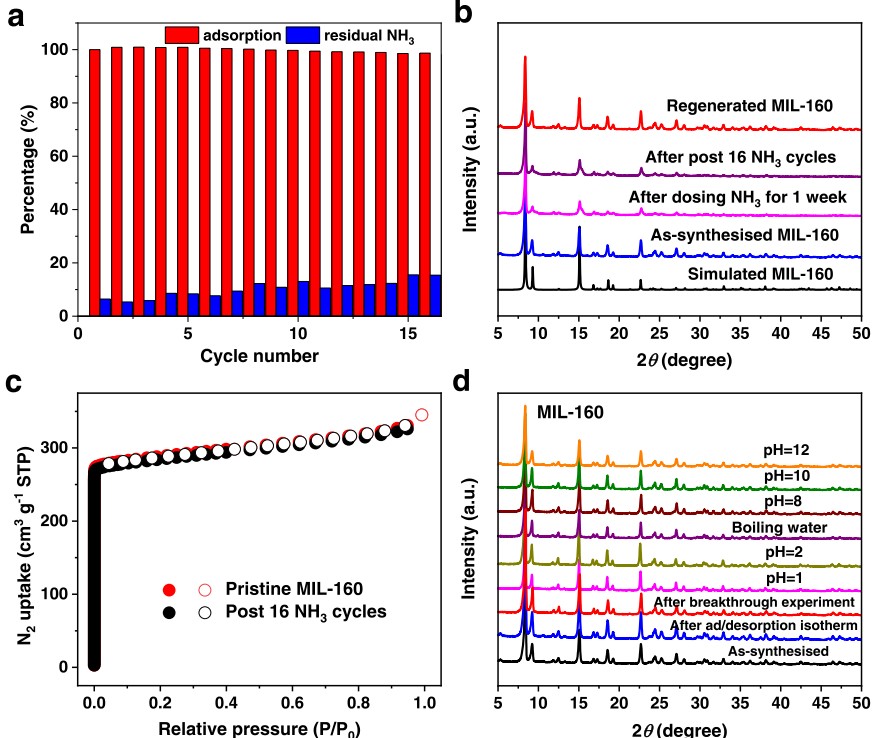

**Fig. 3 Stability data of MIL-160. a** 16 cycles of $NH_3$ adsorption-desorption at 298 K between 0 and 0.2 bar in MIL-160 (pressure-swing conditions) (red bars represents the uptake capacity and blue bars indicate the residual $NH_3$ in the pore upon pressure swing desorption). **b** PXRD patterns of simulated (black), as-synthesised (blue), after dosing $NH_3$ for 1 week (magenta), after post 16 cycles of $NH_3$ adsorption (purple) and regenerated sample (red) for MIL-160. **c** Adsorption-desorption isotherms of $N_2$ at 77 K for pristine MIL-160 (red) and sample regenerated after 16 cycles of $NH_3$ adsorption (black) (solid: adsorption; open: desorption). **d** PXRD patterns of MIL-160 for as-synthesised (black), after $NH_3$ ad/desorption isotherms (blue), breakthrough experiments (red), and samples soaked in solutions with pH = 1 (magenta), 2 (dark yellow), 8 (wine), 10 (olive), 12 (orange) and in boiling water (purple) for 12 h.

is an efficient system for $NH_3$ storage. Moreover, MIL-160 shows high stability with retention of the $NH_3$ capacity during the cyclic adsorption-desorption for at least 16 cycles at 298 K (Fig. 3a). The PXRD patterns of MIL-160 after cycling experiment and after dosing with $NH_3$ for one week (Fig. 3b) show that the Bragg peak at low angle broadens and decreases slightly in intensity, which could be attributed to the distortion of the framework with residual $NH_3$ trapped in the pore, as revealed by ssNMR results (*vide infra*). Residual $NH_3$ can be fully removed by heating at 453 K under dynamic vacuum, and a complete regeneration of the structure (Fig. 3b) with full retention of the porosity (Fig. 3c) is achieved, comparable with other top-performing sorbent materials for $NH_3$[13,16], thus confirming the excellent regenerability of MIL-160 for $NH_3$ storage. MIL-160 also exhibits excellent chemical robustness on adsorption and desorption of $NH_3$ and in breakthrough experiments, as well as in boiling water, acidic and alkaline solutions (Fig. 3d). In contrast, CAU-10-H, Al-fum and MIL-53(Al) undergo structural degradation upon some of these treatments (Supplementary Fig. 9).

**Determination of the binding sites for adsorbed ND₃.** In situ NPD was applied to determine the binding sites of $ND_3$ in MIL-160 (Fig. 4, Supplementary Figs. 10−12 and Supplementary Table 4 − 8). Rietveld refinement of the NPD data of $ND_3$-loaded MIL-160 [MIL-160·$(ND_3)_{0.4}$] reveals two binding sites, denoted as I and II (Fig. 4a and b). Site I (0.202 $ND_3$/Al) exhibits direct binding interactions to the $\mu_2$-OH groups in the pore [$\mu_2$-OH ··· $ND_3$ = 2.36(2) Å], with additional hydrogen bonding [$ND_3$··· $O_{ligand}$ = 2.20(1) Å] and [$ND_3$··· $H − C_{ligand}$ = 2.11(2) Å] and intermolecular interactions with site II [$ND_3$··· $ND_3$ = 2.82(3) Å].

Electrostatic interactions between adsorbed $ND_3$ molecules and the furan rings are also observed [$ND_3$··· $C = C = 2.99(4)$ Å]. Site II (0.220 $ND_3$/Al) is stabilised by hydrogen bonding [$ND_3$··· $O_{ligand}$ = 3.17(2) Å] to the oxygen centre in the furan ring and intermolecular interactions with site I [$ND_3$··· $ND_3$ = 2.82(3) Å]. Three binding sites were observed at higher loading of $ND_3$ in [MIL-160·$(ND_3)_{1.5}$] (Fig. 4c and d); $ND_3$(I) (0.956 $ND_3$/Al) is also stabilised by hydrogen bonding [$\mu_2$-OH ··· $ND_3$ = 2.31(2) Å; $ND_3$··· $O_{ligand}$ = 2.32(4) Å; $ND_3$··· $H − C$ = 2.53(2) Å], supplemented by electrostatic interactions to the furan rings [$ND_3$··· $C = C = 3.60(4)$ Å] as well as intermolecular interactions between site I and site II/site III [$ND_3$··· $ND_3$ = 4.11(2) and 3.79(4) Å]. Site II (0.358 $ND_3$/Al) exhibits hydrogen bonding to the furan ligand [$ND_3$··· $O_{ligand}$ = 2.94(2) Å], together with guest−guest interactions with site I [$ND_3$··· $ND_3$ = 4.11(2) Å]. The $NH_3$ molecules at site III (0.188 $ND_3$/Al) are stabilised by intermolecular interactions to site I [$ND_3$··· $ND_3$ = 3.79(4) Å], which is comparable to the intermolecular distance of $NH_3$···$NH_3$ in solid ammonia[23], suggesting a high packing density of $NH_3$ in MIL-160. Interestingly, in [MIL-160·$(ND_3)_{1.5}$] the hydroxyl groups showed H/D exchange with $ND_3$ molecules at site I in the pore, further confirming the formation of direct host−guest hydrogen bond[10]. The multiple binding sites and efficient packing of $NH_3$ molecules suggest strong host−guest and guest−guest interactions in the framework, consistent with the high value observed for $Q_{st}$ (45 − 63 kJ mol$^{-1}$) for $NH_3$ in MIL-160.

**Analysis of $NH_3$ adsorption in MIL-160 by ssNMR spectroscopy.** The impact of adsorption of $NH_3$ on the framework of MIL-160 was investigated with ssNMR spectroscopy to interrogate

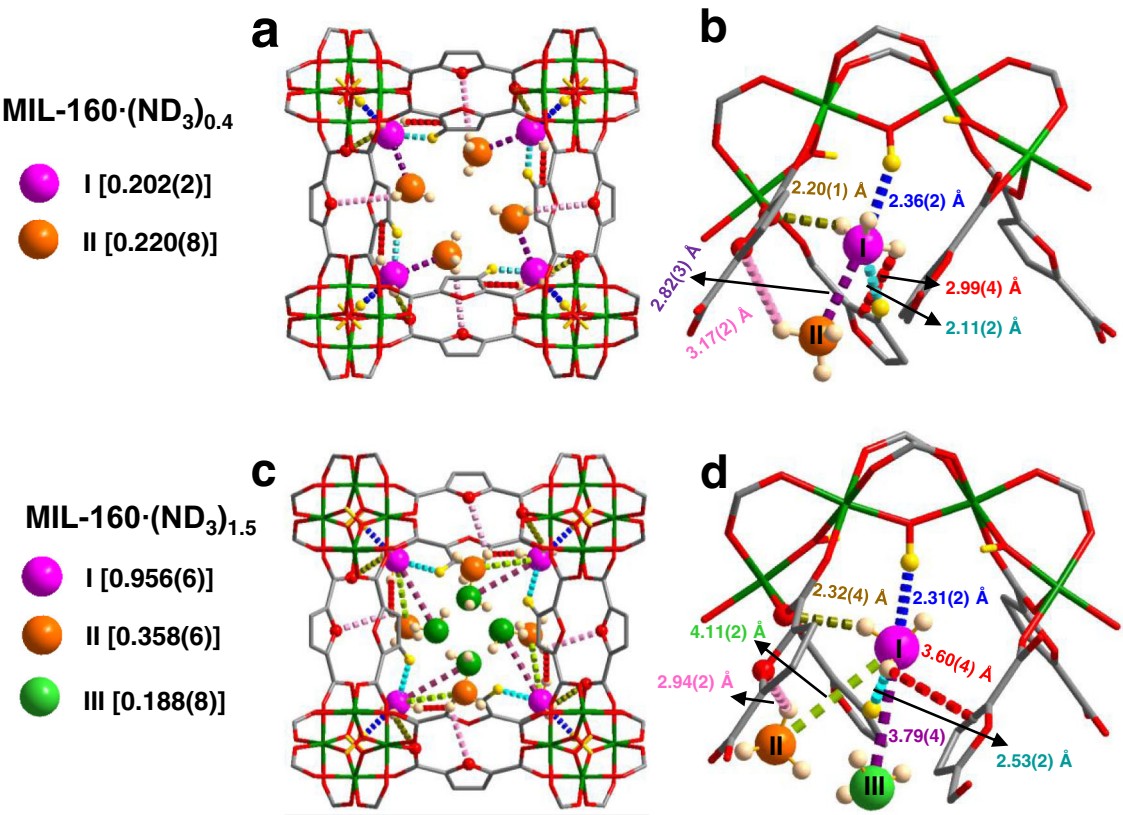

**Fig. 4 Views of crystal structures of ND₃-loaded MIL-160.** Views of the host−guest interactions in ND₃-loaded MIL-160 determined by in situ NPD at 10 K. The occupancy of each site has been converted into ND₃ per Al for clarity. **a** Views of ND₃ in MIL-160·(ND₃)₀.₄ along the *c*-axis and **b** detailed views of host–guest interactions between MIL-160 and ND₃ (Site I: pink, Site II: orange); **c** views of ND₃ in MIL-160·(ND₃)₁.₅ along the *c*-axis and **d** detailed views of host–guest interactions between MIL-160 and ND₃ (Site I: pink; Site II: orange; Site III: green).

any local atomic-scale structural changes[18,24]. The ¹H NMR spectra clearly show the expected presence and absence of NH₃ upon loading and removal of the substrate (Supplementary Fig. 13). Interestingly, after full NH₃ equilibration, an additional NH₃ peak is observed at higher ¹H chemical shift, indicating strong hydrogen bonding and host−guest and guest−guest interactions; this agrees with the NPD analysis (vide supra). Distortions to the environment at the [AlO₆] moieties upon NH₃ uptake was probed using ²⁷Al NMR. The corresponding spectra (Fig. 5a) confirm that the octahedral symmetry is reduced (through an increased $C_Q$) with partial loading of NH₃. Notably, a component related to a distribution of environments begins to appear. Upon equilibration of NH₃ within MIL-160, octahedral moieties of [AlO₆] distorted as the ²⁷Al NMR spectrum displays a line shape characteristic of amorphous octahedral environments, which is further confirmed by the PXRD analysis that shows broadened Bragg peaks and decreased intensity (Fig. 3b). Upon regeneration, the crystalline structure of [AlO₆] moieties is recovered. This ²⁷Al NMR data suggests that notable framework distortion occurs upon NH₃ loading, and this may be attributed to structural "breathing" and/or to adsorption of guest molecules to metal sites[25], as observed for MIL-53(Al)[26]. Breathing has already been shown for MIL-160 when hydrated (i.e., with hydrogen-bonded guest molecules)[27] and this can be linked to changes in observed ²⁷Al NMR parameters[18]. Therefore, it is reasonable to suggest that breathing also occurs for MIL-160(Al) to facilitate uptake of NH₃, but this results in a re-distribution of [AlO₆] geometries locally, which is distinct to the conventional structural phase transition as observed in MIL-53(Al) (Supplementary Fig. 1d).

As suggested above, the primary adsorption sites fill rapidly (Supplementary Fig. 14) and are difficult to regenerate completely

with moderate heating at 150 °C (Supplementary Fig. 8), as the presence of trace NH₃ is still shown in the corresponding ¹H NMR spectrum (Supplementary Fig. 15). A ¹H-¹³C heteronuclear dipolar correlation NMR spectrum (Supplementary Fig. 16) confirms that the NH₃ protons are in close proximity to the carboxylate carbons, indicating their location near the pore corners (Fig. 4) and the hydrogen-bonding of NH₃ to the framework, confirming the results from the NPD (vide supra). The strongly-bound NH₃ required further heating treatment (up to 250 °C) for its complete removal. This treatment caused minor structural degradation of the MOF, as shown through the presence of new peaks in the corresponding ¹³C and ¹H NMR spectrum (Fig. 5b and Supplementary Fig. 17). This degradation is due to the heat treatment alone and not the presence of NH₃ during heating and the NMR chemical shifts of some of these new peaks ($\delta\{^{13}C\}$ ~ 170 ppm (Fig. 5) and $\delta\{^{1}H\}$ ~ 10 ppm (Supplementary Fig. 13) suggest that the regenerated structure contains a small amount of carboxylic acid terminating groups. The amount of this structural degradation has been estimated (using a ratio of the acid ¹³C peak to the carboxylate ¹³C peak: 1:8) to ~13 % of the structure with the retention of porosity and NH₃ capacity. Nevertheless, the overall high capture and storage capacity are remained for regenerated MIL-160.

**In situ spectroscopic analysis of host − guest binding dynamics.** The binding dynamics of MIL-160 upon loading of NH₃ were further investigated by in situ synchrotron infrared (SRIR) microspectroscopy (Fig. 6). The SRIR spectrum (Fig. 6a) of desolvated MIL-160 sample is consistent with the literature data[28]. Upon introduction of 1% NH₃, disappearance of the O−H stretching

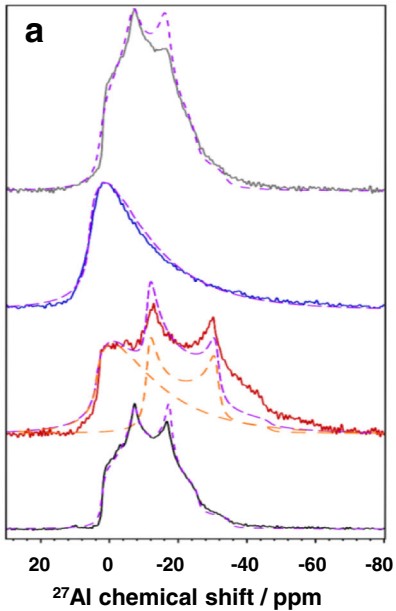 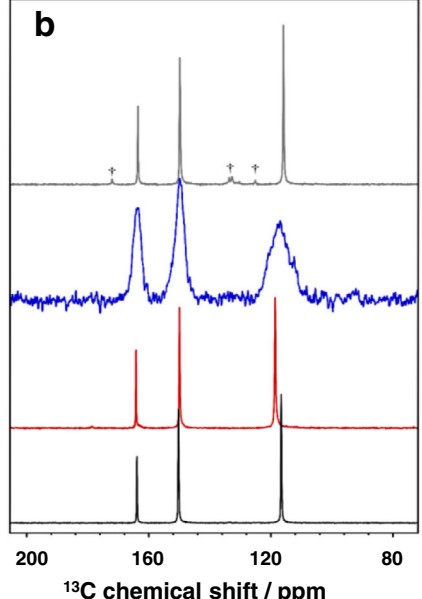

**Fig. 5 Solid-state NMR spectra. a** $^{27}$Al direct excitation and **b** {$^{1}$H-}$^{13}$C cross-polarization MAS NMR spectra of treated MIL-160 samples: pristine (black curve), partial ammonia adsorption (red curve), ammonia saturation after 1 week sealed in a rotor (blue curve) and after active desorption (250 °C for 12 h under dynamic vacuum) (grey curve). Daggers denote peaks arising due to structural decomposition. Simulated $^{27}$Al NMR spectra (purple dashed lines) were produced using the following non-zero parameters: MIL-160 $C_Q = 5.2$ MHz, $\eta_Q = 0.4$, $\delta_{iso} = 3.8$ ppm, NH$_3$-MIL-160' $C_Q = 4.8$ MHz, $\delta_{iso} = 4.0$ ppm (Gaussian Isotropic Distribution Model) and $C_Q = 5.5$ MHz, $\eta_Q = 0$, $\delta_{iso} = -5.4$ ppm, NH$_3$-MIL-160 $C_Q = 4.8$ MHz, $\delta_{iso} = 6.0$ ppm, NH$_3$-MIL-160$_{250}$ $C_Q = 5.2$ MHz, $\eta_Q = 0.46$, $\delta_{iso} = 4.0$ ppm.

band at 3686 cm$^{-1}$ was observed (Fig. 6b), further confirming the strong host−guest interaction between NH$_3$ molecules and the framework. The band at 1655 cm$^{-1}$ is assigned to the asymmetric stretching vibrations of the carboxylate groups. Upon loading with NH$_3$ (1 − 100%), the peak shows red shifts to 1644 cm$^{-1}$ ($\Delta = 11$ cm$^{-1}$) (Fig. 6c). The characteristic band at 1574 cm$^{-1}$ is assigned to the C=C bond stretching mode, and the peak at 780 cm$^{-1}$ to the out-of-plane deformation vibrations of C−H bonds in the furan rings. On dosing with NH$_3$, the red shift ($\Delta = 7$ cm$^{-1}$) of the peak at 1574 cm$^{-1}$ reflects the presence of NH$_3 \cdots$ C=C interactions (Fig. 6d). The peaks in the range of 1000−1250 cm$^{-1}$ can be attributed to the asymmetric and symmetric stretching vibrations of the C−O−C in the furan rings. Specifically, the peak at 1243 cm$^{-1}$ shows a red shift and broadens, while the band at 1013 cm$^{-1}$ both reduces in intensity and broadens as NH$_3$ loading increases (Fig. 6e). The emergence of a new band at 1101 cm$^{-1}$ can be assigned to the N−H wagging upon NH$_3$ adsorption. When the loading of NH$_3$ increases to 40%, the peak broadens indicative of a more complex binding environment, consistent with the NPD data (*vide supra*). These results are consistent with the interactions of ND$_3 \cdots$ O$_{ligand}$. Furthermore, the blue shift of the C-H deformation band to 785 cm$^{-1}$ ($\Delta = 5$ cm$^{-1}$) is observed, again consistent with the presence of interactions between NH$_3$ and furan rings (Fig. 6f). Upon regeneration, the entire spectrum of the framework returns to that of the orginal activated material, confirming the high structural robustness of MIL-160.

## Conclusions

We report the crucial effects of functional groups (e.g., μ$_2$-OH), pore geometry and structural flexibility on the development of Al-based MOF materials for efficient capture and storage of NH$_3$. At 298 K and 1.0 bar, NH$_3$ uptakes follow the order of MIL-160 (12.8 mmol g$^{-1}$) > CAU-10-H (10.0 mmol g$^{-1}$) > Al-fum (8.9 mmol g$^{-1}$) > MIL-53(Al) (3.0 mmol g$^{-1}$). The suitable pore size, anchored μ$_2$-OH, and the O-heteroatom of the furan linker

within the channel of MIL-160 enable strong interactions with NH$_3$ molecules, thus promoting the excellent adsorption of NH$_3$ at both low and high pressure. The in situ NPD, synchrotron IR and ssNMR spectroscopy reveal the adsorption mechanism and interaction with μ$_2$-OH groups in the pores and distortion of the [AlO$_6$] moieties upon NH$_3$ uptake. Considering the advantages of the high NH$_3$ affinity and uptakes, and high stability, MIL-160 has a great potential in practical application as a robust sorbent for NH$_3$.

## Methods and characterisation

**Synthesis of Al-MOFs.** MIL-160, CAU-10-H, Al-fum, and MIL-53(Al) were synthesised according to reported methods with small modifications[18–21]. MIL-160 was prepared by reaction of NaOH (0.08 g, 2.0 mmol), H$_2$fdc (0.15 g, 1.0 mmol), H$_2$O (15 mL) and AlCl$_3$·6H$_2$O (0.24 g, 1.0 mmol) under reflux at 378 K for 12 h. The product was collected by filtration and washed with DMF and H$_2$O, and then exchanged 3 days with acetone.

CAU-10-H was prepared by reaction of Al$_2$(SO$_4$)$_3$·18H$_2$O (0.4 g, 0.6 mmol), $m$-H$_2$bdc (0.1 g, 0.6 mmol), H$_2$O (4 mL) and DMF (1 mL) in a Teflon-lined stainless-steel autoclave (408 K, 12 h). The product was collected by filtration and washed with DMF and H$_2$O, and then exchanged for 3 days with acetone.

Al-fum was prepared by reaction of Al$_2$(SO$_4$)$_3$·18H$_2$O (0.4 g, 0.6 mmol), NaOH (0.15 g, 3.6 mmol) and H$_2$fum (0.14 g, 1.2 mmol) in H$_2$O (10 mL) with sonication for 5 min. The solution was transferred into a 38 mL pressure tube which was heated at 363 K for 2 h. The product was collected by filtration, washed with H$_2$O, and then exchanged 3 days with acetone.

MIL-53(Al) was prepared by reaction of Al(NO$_3$)$_3$·9H$_2$O (1.31 g, 3.5 mmol), $p$-H$_2$bdc (0.29 g, 1.7 mmol) and H$_2$O (15 mL) in a Teflon-lined stainless-steel autoclave at 483 K for 3 days. After cooling, the product was collected by filtration and washed with H$_2$O. The dried white powder was then calcined in a muffle-furnace with the air flow at 603 K for 3 days to remove the incorporated $p$-H$_2$bdc from the pores, and then stored under acetone.

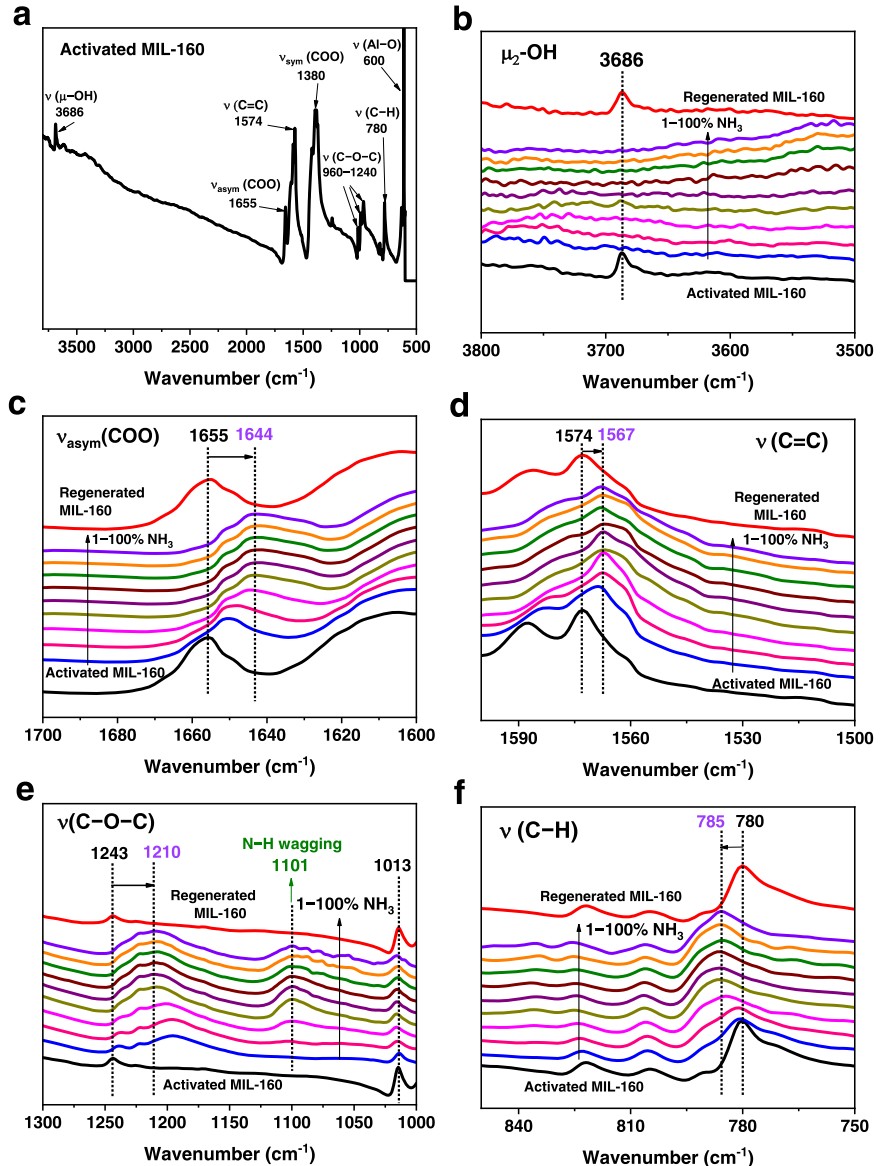

**Fig. 6 In situ synchrotron IR spectra. a** In situ synchrotron IR spectra for activated MIL-160; In situ synchrotron IR spectra for MIL-160 as a function of adsorption of $NH_3$ (diluted in dry $N_2$) and after regeneration under a dry $N_2$ flow at 10 mL min$^{-1}$ at 423 K for 2 h: **b** 3800-3500 cm$^{-1}$, **c** 1700-1600 cm$^{-1}$, **d** 1600-1500 cm$^{-1}$, **e** 1300-1000 cm$^{-1}$, **f** 850-750 cm$^{-1}$. Activated MIL-160 (black), 1% $NH_3$-loaded MIL-160 (blue), 2% $NH_3$-loaded MIL-160 (pink), 5% $NH_3$-loaded MIL-160 (magenta), 10% $NH_3$-loaded MIL-160 (dark yellow), 20% $NH_3$-loaded MIL-160 (purple), 40% $NH_3$-loaded MIL-160 (wine), 60% $NH_3$-loaded MIL-160 (olive), 80% $NH_3$-loaded MIL-160 (orange), 100% $NH_3$-loaded MIL-160 (violet), regenerated MIL-160 (red).

**General characterisation**. Powder X-ray diffraction patterns were collected using a Philips X'pert X-ray diffractometer (40 kV and 30 mA) using Cu Kα radiation ($\lambda = 1.5406$ Å). The pore size, and surface areas were obtained from $N_2$ isotherms recorded on a 3-flex (Micrometrics) instrument at 77 K. Thermogravimetric analysis was conducted on a TA Instrument Q600 under air flow of 100 mL min$^{-1}$.

**Ammonia temperature-programmed desorption (TPD)**. Temperature-programmed desorption of ammonia ($NH_3$-TPD) with a Quantachrome Autosorb-1 equipped with a thermal conductivity detector (TCD) was performed to assess the affinity of $NH_3$ in MIL-160 framework. Typically, 80 mg of sample was pre-treated in a helium stream (30 mL min$^{-1}$) at 150 °C for 10 h. The adsorption of $NH_3$ was carried out at 50 °C for 1 h. The sample was flushed with helium at 100 °C for 2 h to remove physisorbed $NH_3$ from the sample surface. The TPD profile was recorded at a heating rate of 10 °C min$^{-1}$ from 100 to 300 °C.

**Gas adsorption and breakthrough experiments**. Measurements of static adsorption isotherms (0−1.0 bar) for $NH_3$ were undertaken on an IGA gravimetric sorption analyser (Hiden Isochema, Warrington, UK) under ultra-high vacuum with the temperature controlled using a programmed water bath. Research-grade $NH_3$ was purchased from BOC and used as received. Acetone exchanged samples were loaded into the IGA system and degassed at 423 K and $1 \times 10^{-6}$ mbar for 10 h to give a desolvated material of typical mass ca. 30 mg. For the cycling experiments, the pressure of $NH_3$ was increased from vacuum ($1 \times 10^{-8}$ mbar) to 0.2 bar and the uptake recorded. The pressure was then reduced to regenerate the sample without heating. Dynamic breakthrough experiments were conducted on a Hiden Isochema IGA-003 with ABR attachments and a Hiden Analytical mass

spectrometer by using a fixed-bed tube packed with 750 mg of powder. The sample was heated at 423 K under a flow of dry He for 12 h for activation, and then cooled to 298 K. The dynamic breakthrough experiments were collected at a concentration of 1000 ppm $NH_3$ (diluted in He) at the total flow rate of 25 mL min$^{-1}$. The concentration of $NH_3$ in the outlet was determined by mass spectrometry and compared with the inlet concentration $C_0$, where $C/C_0 = 1$ indicates complete breakthrough. To determine the dynamic adsorption capacity, the uptake of each component ($n_m$) was calculated based on the breakthrough curves by the following equation:

$$V_m = \frac{\int_0^t v_{gas\,out} \mathrm{d}t - V_{dead}}{W_{MOF}} \quad (1)$$

$$n_m = \frac{PV_m}{RT} \quad (2)$$

where $v_{\mathrm{gas\,out}}$ is the flow rate of the target gas with the unit of mL min$^{-1}$, $V_{\mathrm{dead}}$ is the dead volume of the system (mL), $W$ represents the mass of sample packed in the breakthrough bed (g). $t$ is the retention time for the specific gas (min), $P$ is pressure (kpa), R is gas constant, and $T$ is the measurement temperature (K).

**In situ neutron powder diffraction experiments**. Neutron powder diffraction (NPD) experiments for $ND_3$-loaded MIL-160 were undertaken on the WISH diffractometer at the ISIS Facility at Rutherford Appleton Laboratory (UK). The instrument has a solid methane moderator providing a high flux of cold neutrons with a large bandwidth, transported to the sample via an elliptical guide. The divergence jaws of the WISH system allow tuning of the resolution according to the need of the experiment; in this case, it was setup in high resolution mode. The WISH detectors are 1 m long, 8 mm diameter pixelated $^3$He tubes positioned at 2.2 m from the sample and arranged on a cylindrical locus covering a $2\theta$ scattering angle of $10 - 170°$. To reduce the background from the sample environment, WISH was equipped with an oscillating radial collimator that defines a cylinder of radius of approximately 22 mm diameter at 90º scattering angle. The sample of desolvated MIL-160 was loaded into a cylindrical vanadium sample container with an indium vacuum seal connected to a gas handling system. The sample was degassed at $1 \times 10^{-7}$ mbar and at 373 K for 12 h with He flushing to remove any remaining trace of guest molecules. The sample was dosed with $ND_3$ using the volumetric method after being warmed to room temperature (298 K) to ensure that the gas was well dispersed. A certain amount of $ND_3$ was dosed into the vanadium holder containing MIL-160. The ratio of $ND_3$ to the MOF was calculated through the difference on the partial pressure of the $ND_3$ in the buffer container (500 mL) before and after dosing, based on the equation $PV = nRT$, where the $T$ is 298 K, R is the gas constant and $V$ is the dead volume of the system (mL). Data collection for desolvated MIL-160 and two subsequent loadings of $ND_3$ (0.4 and 1.5 $ND_3$ molecules per metal site) were performed while the temperature was controlled using a He cryostat ($10 \pm 0.2$ K).

Rietveld structural refinements were carried out on the NPD data using Bruker-AXS Topas (V5.0). The structure of desolvated MIL-160 was established based on the reported crystallographic structure and subsequently refined against the NPD pattern for activated MIL-160[18]. Soft restraints were applied on bond lengths and bond angles within the furan rings and carboxylates to keep the molecule integrity. Isotropic displacement parameters ($U_{iso}$) were used for all non-H atoms, where the riding model was used for the hydrogen's displacement parameters. Upon loading of $ND_3$, obvious changes in peak intensity were observed indicating the successful adsorption of molecules in the bare framework.

The $ND_3$ molecule was modelled as rigid body with fixed bond lengths and angles from DFT-optimised molecule geometry (B3LYP-D$_3$, 6-31 G**, Gaussian09). The position of the primary $ND_3$ molecule was extracted from the difference Fourier map. The initial positions of other loaded $ND_3$ were guessed from simulated annealing via the Auto_T macro in TOPAS for molecules placed in general positions, where precise locations were obtained by subsequent refinement. Two binding sites were located with a total occupancy reaching 0.4 $ND_3$ per Al atom for the low-loading sample. An additional adsorption site filling the pore was obtained from the NPD pattern of high-loading sample with a total occupancy around 1.5 $ND_3$ per metal atom.

For the low-loading sample, no positional restraint was applied, while the fractional $x$ coordinate of the N atom of site III $ND_3$ in the high loading structure was found to vibrate closely around a special position and was consequently fixed. All translations and rotations of $ND_3$ molecules were freed with no restraints allowing precise atomic positions of D of $ND_3$ to be determined. Isotropic displacement parameters were used for all N atoms, where the parameter for hydrogen was defined similarly with a riding model. Refining the H of the hydroxy group (−OH) on the framework as a combination of −OH and −OD in the high loading structure gave lower R-factors than purely −OH. This indicated the presence of H–D exchange between active framework hydroxyl groups and site I $ND_3$ molecules as the result of strong host−guest interactions. This exchange is apparent in the refinement of NPD data owing to the significant difference of neutron scattering length ($b_c$, in fm) of proton (H, −3.74) and deuteron (D, 6.67).

**Solid-state nuclear magnetic resonance experiments**. A 400 MHz Bruker Advance III spectrometer (9.4 T) was used at ambient temperature with a 4 mm HFX probe and a MAS frequency of 12 kHz. Samples were activated (423 K for 10 h under dynamic vacuum) and packed into 4 mm outer diameter zirconia rotors under inert conditions. The various sample treatments were applied to the sample in situ in the rotor. Spectral simulations and fitting were performed in the solid lineshape analysis (SOLA) module v2.2.4 in Bruker TopSpin v4.0.9 for crystalline models and in DMFit[29] for Gaussian isotropic distribution models. $^1$H and $^{13}$C chemical shifts are given with respect to TMS (0 ppm) and $^{27}$Al chemical shifts are referenced to a 1.1 mol/kg $Al(NO_3)_3$ in $D_2O$ solution. More information on specific experimental details may be found in Supplementary Note 2 in the supporting information.

**In situ synchrotron infrared micro-spectroscopy experiments**. In situ synchrotron infrared micro-spectroscopy experiments were carried out at multimode infrared imaging and micro spectroscopy (MIRIAM) beamline at the Diamond Light Source, UK. Measurements were performed using a Bruker Vertex 80 V FTIR equipped with a mid-infrared LN$_2$-cooled MCT (Mercury Cadmium Telluride) detector and the Diamond Light Source synchrotron as an IR source. Spectra were collected in the range $4000 - 400$ cm$^{-1}$ and aperture size at the sample of approximately $20 \times 20$ μm. A microcrystalline powder of MIL-160 was scattered onto a 0.5 mm thick ZnSe infrared window and placed within a Linkam FTIR 600 gas-tight sample cell equipped with 0.5 mm thick ZnSe windows, a heating stage and gas inlet and outlet. Ultrapure $N_2$ and anhydrous $NH_3$ gases were used as supplied from the cylinder. The gases were dosed volumetrically to the sample cell using mass flow controllers, and the total flow rate was maintained at 100 mL min$^{-1}$ for all experiments. The exhaust from the cell was directly vented to an extraction system and the total pressure in the cell was therefore 1.0 bar for all experiments. The sample was desolvated under a flow of dry $N_2$ at

$100\,mL\,min^{-1}$ and 423 K for 5 h, and was then cooled to 298 K under a continuous flow of $N_2$. Dry $NH_3$ was then dosed as a function of partial pressure, maintaining a total flow of $100\,mL\,min^{-1}$ with $N_2$ as a balance gas. The MOF sample was then regenerated with a flow of dry $N_2$.

## Data availability

Supplementary information contains additional synthesis procedures, characterisation, and analysis of crystal structures and adsorption results. Crystal data are deposited at Cambridge Crystallographic Data Centre (CCDC), under deposition numbers CCDC 2219217 (Supplementary Crystallographic Data 1: CIF for bare MIL-160), 2219215 [Supplementary Crystallographic Data 2: CIF for MIL-160 · $(ND_3)_{0.4}$], 2219216 [Supplementary Crystallographic Data 3: CIF for MIL-160 · $(ND_3)_{1.5}$]. These data can be obtained free of charge from The Cambridge Crystallographic Data Centre via https://www.ccdc.cam.ac.uk/structures/. The data that support the conclusion of this study are presented in the supplementary information and available from the corresponding authors on reasonable request.

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

## Acknowledgements

We thank EPSRC (EP/I011870, EP/V056409), Royal Society, and University of Manchester funding. This project has received funding from the European Research Council (ERC) under the European Union's Horizon 2020 research and innovation programme (grant agreement No 742401, NANOCHEM). We are grateful to ISIS facility and Diamond Light Source for access to Beamlines WISH and B22, respectively. L.G., M.H. and W.H. thank the China Scholarship Council (CSC) for funding.

## Author contributions

L.G. and W.H. synthesised and characterised the MOF samples, measured and analysed adsorption isotherms and the breakthrough data. J.H. and D.L. collected and analysed the ssNMR data. M.H., J.L., D.C., M.F., S.S., P.M., Y.C. and L.G. collected and analysed the NPD data. W.L., X.Z., J.L., M.K.-J., M.D.F. and L.G. collected and analysed SRIR data. L.G. and Z.Z. collected the TPD data. L.G., J.H., M.H., D.L., M.S. and S.Y prepared the manuscript. M.S. and S.Y. jointly supervised this work.

## Competing interests

The authors declare no competing interests.
