## [Peer Review File · Communications Chemistry]

Reviewers' comments:

Reviewer #1 (Remarks to the Author):

In the paper, the authors report Al-based MOFs, MIL-160, CAU-10-H, Al-fum, MIL-53(Al) for efficient capture and storage of ammonia. This paper was well written and the preferred NH₃ adsorption domain and mechanism in MIL-160 were fully characterized through the appropriate characterization tools including in situ NPD, sIR and ssNMR spectroscopy. Although NH₃ adsorption performance and packing densities were not exceptionally high compared to other reported MOF materials, the binding sites and adsorption mechanism are well documented. I would recommend this study for publication in Communications Chemistry. Here are some minor comments to be considered.

-Ammonia temperature-programmed desorption (TPD) data should be included to assess the affinity of NH₃ in the framework depending on the adsorption sites. In addition, the primary adsorption sites are difficult to regenerate completely with moderate heating (150 °C), confirming the presence of trace NH₃ in the framework. Meanwhile, further heating treatment (up to 250 °C) exhibited complete removal of NH₃, but it caused minor structural degradation of the MOF. Therefore, the regeneration condition with proper temperature should be determined through TPD analysis.

Reviewer #2 (Remarks to the Author):

This is a very good paper, and one that certainly fits with the special issue topic.

Ammonia adsorption is challenging due to the nature of the gas, and is especially so if cycling of the material is required for any length of time. The trick is to balance the gas-material interactions - strong enough for high adsorption but not too strong that the corrosive nature of the gas becomes a major problem. To understand this we need excellent characterisation of the gas inside the pores of the solid, and this paper shows how this can be done with neutron diffraction, solid state NMR and spectroscopy. I am in favour of publication but have the following questions/comments that should be addressed.

1. Clearly there is the potential for breakdown of the material under temperature swing conditions and even under pressure swing one can see considerable loss of capacity of MIL-160. Can the authors comment a little more critically on this. For real world applications this would be an issue limiting the lifetime of the material and deserves to be critically assessed.
2. The paper mentions breathing of the MIL-160, but only in the NMR section. Breathing in MIL-53 should be very easy to spot using diffraction - was there any evidence in the diffraction experiments? If it is not visible then the authors should say so.
3. There is a large change in the ²⁷Al NMR spectra after leaving the material in contact with NH₃ for a week. The authors say that "short-range order seems to disappear as the ²⁷Al NMR spectrum displays a line shape characteristic of amorphous octahedral environments". I have no idea why this means - amorphous materials are usually associated with loss of long range order. What does the XRD pattern look like after a week in contact with NH₃? Clearly there are irreversible changes going on as seen in the spectra which probably account for the loss of capacity on cycling. This type of change is really important to study in a situation like this, and while it is not necessary for this paper is something that should be looked at in much greater detail.

4. The neutron diffraction section mentions the dosing of NH₃ using the 'volumetric method'. I have no idea what that means either - please expand.

5. The figure showing the location of the NH₃ species is misleading for students. Unless there is evidence to the contrary sites ii and iii would likely not be filled simultaneously because of their occ. Therefore marking an internuclear distance makes no sense. Students often get confused by this. I would expand the figure to show two cages where the ii and iii are only shown in one of the cages. Site i, however, with an occupancy of nearly 1 would be present in both cages. This would clean up the figure a little as the marked distances would only be the ones that are present in the real crystal. The internuclear distance (say i to ii) is a real one - how does this compare to NH₃ NH₃ distances in say, solid ammonia? It doesn't look like there is H-bonding between these two sites?

Overall, the paper does report some excellent advanced characterisation and so is suitable for the call and should be published. One can always be picky about the temperature of the neutron experiment (10 K) compared to real life - it would have been nice (if expensive) to also collect data at room T to at least see whether the N atoms were in the same place - but one cannot use such resources optimally all the time because of accessibility so that is not a real criticism - just something to ponder for future neutron work.

Reviewer #3 (Remarks to the Author):

The manuscript deals with the capture and storage of ammonia in different types of aluminum-based MOFs. This process is of high importance for the community and can create severe problems for the given adsorption system. Four MOFs are compared with focus on MIL-160 which performs best over all four. Characterization of the material is done in multiple ways. The manuscript topic and quality is clearly valid for publication. I have, however, some smaller and larger concerns that should be considered before publication.

My main concern is that several characterization techniques are used but, probably because of the page limitation of the journal, the discussion is in several positions rather short. More details would be needed. For the same reason most of the results are published in supplementary information. To my view it would be much better if the manuscript is expanded significantly even if that means that it should be published in another journal. As it is now, it's like a data collection that is briefly discussed.

More specific comments:

- it is really complicated to follow the discussion because figures are not embedded within the text
- Fig. S1: the synthesized MOFs show in part strong differences to the simulated profiles. Major reflections are there but also many other, there is no comment on this in the text
- p.4 about middle: ... degradation under these some of these conditions...
- just below: why Suppl Table 6 not 1?
- Suppl. Fig. 16: notation of samples not intuitive; should be also add some in caption; furthermore, NH₃-MIL-160 contains more peaks than just the one at higher shift. Also spectra should be plotted in a scaled fashion (on a rather absolute scale), it is not trivial to differentiate signals of the MOF and NH₃. Analysis seems to be not detailed enough; not all spectra of the different samples are shown
- Suppl. Fig. 17: nothing is said about ¹³C spectra, signal assignment?; ¹H spectra are repeated from Fig. 16, why?
- Suppl. Fig. 19: no clear analysis of spectra
- Suppl. Fig. 20: no link in manuscript; no F1 axis
- there is no link in the manuscript referring to Suppl. Tables 1-5 & 7-8
- Methods and characterization: some parts repeated in/from SI

We thank the reviewers for their constructive comments, and our responses to the points raised by the reviewers are given below in *bold italics*. The changes have now been highlighted **in yellow** for the revised manuscript and SI.

Reviewer #1 (Remarks to the Author):

In the paper, the authors report Al-based MOFs, MIL-160, CAU-10-H, Al-fum, MIL-53(Al) for efficient capture and storage of ammonia. This paper was well written and the preferred NH₃ adsorption domain and mechanism in MIL-160 were fully characterized through the appropriate characterization tools including in situ NPD, sIR and ssNMR spectroscopy. Although NH₃ adsorption performance and packing densities were not exceptionally high compared to other reported MOF materials, the binding sites and adsorption mechanism are well documented. I would recommend this study for publication in Communications Chemistry. Here are some minor comments to be considered.

Ammonia temperature-programmed desorption (TPD) data should be included to assess the affinity of NH₃ in the framework depending on the adsorption sites. In addition, the primary adsorption sites are difficult to regenerate completely with moderate heating (150 °C), confirming the presence of trace NH₃ in the framework. Meanwhile, further heating treatment (up to 250 °C) exhibited complete removal of NH₃, but it caused minor structural degradation of the MOF. Therefore, the regeneration condition with proper temperature should be determined through TPD analysis.

Revised. Thank you for this constructive advice. Ammonia temperature-programmed desorption (TPD) has been conducted and results added (Supplementary Figure 8).

Reviewer #2 (Remarks to the Author):

This is a very good paper, and one that certainly fits with the special issue topic.

Ammonia adsorption is challenging due to the nature of the gas, and is especially so if cycling of the material is required for any length of time. The trick is to balance the gas-material interactions - strong enough for high adsorption but not too strong that the corrosive nature of the gas becomes a major problem. To understand this we need excellent characterisation of the gas inside the pores of the solid, and this paper shows how this can be done with neutron diffraction, solid state NMR and spectroscopy. I am in favour of publication but have the following questions/comments that should be addressed.

1. Clearly there is the potential for breakdown of the material under temperature swing conditions and even under pressure swing one can see considerable loss of capacity of MIL-160. Can the authors comment a little more critically on this. For real world applications this would be an issue limiting the lifetime of the material and deserves to be critically assessed.

Revised. Additional results on the stability have been moved from SI to the manuscript (Figure 3) and the discussion on the stability for all four Al-MOFs highlighted in yellow and moved to the section on "Regeneration and stability test".

2. The paper mentions breathing of the MIL-160, but only in the NMR section. Breathing in MIL-53 should be very easy to spot using diffraction - was there any evidence in the diffraction experiments? If it is not visible then the authors should say so.

Revised. The PXRD patterns of activated and NH₃-loaded MIL-53(Al) have been shown in Supplementary Figure 1d, which confirmed the breathing phenomenon in MIL-53(Al). The results and discussion have been added in the main manuscript.

3. There is a large change in the ²⁷Al NMR spectra after leaving the material in contact with NH₃ for a week. The authors say that "short-range order seems to disappear as the ²⁷Al NMR spectrum displays a line shape characteristic of amorphous octahedral environments". I have no idea why this means - amorphous materials are usually associated with loss of long range order. What does the XRD pattern look like after a week in contact with NH₃? Clearly there are irreversible changes going on as seen in the spectra which probably account for the loss of capacity on cycling. This type of change is really important to study in a situation like this, and while it is not necessary for this paper is something that should be looked at in much greater detail.

Revised. We apologise for this typo. This has now amended as "Upon equilibration of NH₃ within MIL-160, octahedral moieties of [AlO₆] distorted as the ²⁷Al NMR spectrum displays a line shape characteristic of

amorphous octahedral environments, whereas long-range order remained as suggested by PXRD pattern (Figure 3b)."

4. The neutron diffraction section mentions the dosing of NH₃ using the '**volumetric method**'. I have no idea what that means either - please expand.

Revised. More experimental details have been added in the Methods and Characterisation.

5. The figure showing the location of the NH₃ species is misleading for students. Unless there is evidence to the contrary sites ii and iii would likely not be filled simultaneously because of their occ. Therefore marking an internuclear distance makes no sense. Students often get confused by this. I would expand the figure to show two cages where the ii and iii are only shown in one of the cages. Site i, however, with an occupancy of nearly 1 would be present in both cages. This would clean up the figure a little as the marked distances would only be the ones that are present in the real crystal. The internuclear distance (say i to ii) is a real one - how does this compare to NH₃-NH₃ distances in say, solid ammonia? It doesn't look like there is H-bonding between these two sites?

Revised. Thanks for this constructive advice. (1) Figure 4 and Supplementary Table 7 have been updated after removing the internuclear distance of site II and III. (2) The N-N distance in solid ammonia of phase I ranging from 3.17 to 3.94 Å (Phys. Rev. Lett. 76,74 (1996)). There was no H-bonding observed between site I and II/III, and the distance of N_{site I}-N_{site II} (4.11 Å) and N_{site I}-N_{site III} (3.79 Å) are comparable to solid ammonia, suggesting high NH₃ packing density in MIL-160. This has been fully revised in the manuscript.

Overall, the paper does report some excellent advanced characterisation and so is suitable for the call and should be published. One can always be picky about the temperature of the neutron experiment (10 K) compared to real life - it would have been nice (if expensive) to also collect data at room T to at least see whether the N atoms were in the same place - but one cannot use such resources optimally all the time because of accessibility so that is not a real criticism - just something to ponder for future neutron work.

We thank for the advice and will plan for future investigation on the variable temperature NPD experiments.

Reviewer #3 (Remarks to the Author):

The manuscript deals with the capture and storage of ammonia in different types of aluminum-based MOFs. This process is of high importance for the community and can create severe problems for the given adsorption system. Four MOFs are compared with focus on MIL-160 which performs best over all four. Characterization of the material is done in multiple ways. The manuscript topic and quality is clearly valid for publication. I have, however, some smaller and larger concerns that should be considered before publication.

My main concern is that several characterization techniques are used but, probably because of the page limitation of the journal, the discussion is in several positions rather short. More details would be needed. For the same reason most of the results are published in supplementary information. To my view it would be much better if the manuscript is expanded significantly even if that means that it should be published in another journal. As it is now, it's like a data collection that is briefly discussed.

Revised. Key Supplementary Figures have been moved to the manuscript and additional discussion added.

More specific comment:

- it is really complicated to follow the discussion because figures are not embedded within the text

Revised. These are now embedded within the text.

- Fig. S1: the synthesized MOFs show in part strong differences to the simulated profiles. Major reflections are there but also many other, there is no comment on this in the text.

Revised. Additional analysis and a reference for Al-fum MOF have been added.

- p.4 about middle: ... degradation under these some of these conditions...

- just below: why Suppl Table 6 not 1?

Revised. We apologise for the typo. This should be Supplementary Table 1.

- Suppl. Fig. 16: notation of samples not intuitive; should be also add some in caption; furthermore, NH₃-MIL-160 contains more peaks than just the one at higher shift. Also spectra should be plotted in a scaled fashion

(on a rather absolute scale), it is not trivial to differentiate signals of the MOF and NH₃. Analysis seems to be not detailed enough; not all spectra of the different samples are shown.

Revised. Supplementary Figure 16 has been updated, referred as Supplementary Figure 14; The description has been updated as “An additional NH₃ peak was observed at higher shift in NH₃-MIL-160..” ; The NMR study was mostly focused on the robust material MIL-160 that shows the best performance in NH₃ adsorption in this study.

- Suppl. Fig. 17: nothing is said about ¹³C spectra, signal assignment? ¹H spectra are repeated from Fig. 16, why?

Revised. Supplementary Figure 17 has been updated, referred as Supplementary Figure 15; ¹H spectra was repeated for ease of comparison, this has now been made clearer in the associated caption.

- Suppl. Fig. 19: no clear analysis of spectra this.

Revised. Additional analysis has been added with a text reference in the main manuscript.

- Suppl. Fig. 20: no link in manuscript; no F1 axis.

Revised. Suppl. Fig. 20 was originally included to provide supplementary evidence for the data fitting, but it is clear that the fits are unambiguous so this has now been removed.

- there is no link in the manuscript referring to Suppl. Tables 1-5 & 7-8.

Revised. All supplementary figures and tables have been cited in the main manuscript.

- Methods and characterization: some parts repeated in/from SI.

Revised. The repeated information has been removed from SI.

REVIEWERS' COMMENTS:

Reviewer #1 (Remarks to the Author):

The authors added the new data I requested and I recommend publication of this manuscript.

Reviewer #2 (Remarks to the Author):

Generally, the authors have done a good job in answering the questions of the reviewers - I thought they were a bit brief in answering some of the excellent points of Referee 3 (which was not me).

Perhaps my only issue is the categorical statement regarding the stability of the material. After 16 cycles it looks like they have lost significant capacity (especially the blue bars for adsorption). Depending on the application it looks like a steady loss of working capacity. Therefore it isn't perfectly stable under these conditions (but might be better than other MOFs). Consider rewording further. But otherwise this is fine for publication

Reviewer #3 (Remarks to the Author):

In the revised manuscript, the authors addressed all points raised by the different referees. Moving some figures from the supplementary to the main text is clearly beneficial for the article.

There is just one point remaining that - after changing according to a referee's question - is still confusing.

Bottom of page 8, the changed text to the 27Al spectra. First, I think the sentence structure is wrong, at least I don't understand the grammar. Second, the content is confusing me. If there are distorted octahedral moieties, then there is no long range order which is shown by the PXRD. So NMR and PXRD don't coincide. Was the PXRD done from the same material that was left in the MAS rotor for one week or a similar sample stored NH₃ for a week? In the latter case it might be that different samples are compared with NMR and PXRD. Otherwise I would be very surprised. The NMR spectrum shows that there is practically no 'crystalline' signal, so the overall content has to be very small. The PXRD shows just slight broadening of the peaks. Is there some indication that the 'intensity' is much lower in the PXRD compared to the other samples so that PXRD just 'sees' the tiny amount of crystalline parts remaining? From my experience, the former case is quite often occurring... The authors should comment on this, even if they write that they cannot explain the confusing results between NMR and PXRD. With that, the manuscript should be ready to go.

We thank the reviewers for their constructive comments, and our responses to the points raised by the reviewers are given below in *bold italics*. The changes have now been highlighted **in yellow** for the revised manuscript and SI.

Reviewer #1 (Remarks to the Author):

The authors added the new data I requested and I recommend publication of this manuscript.

Reviewer #2 (Remarks to the Author):

Generally, the authors have done a good job in answering the questions of the reviewers - I thought they were a bit brief in answering some of the excellent points of Referee 3 (which was not me).

Perhaps my only issue is the categorical statement regarding the stability of the material. After 16 cycles it looks like they have lost significant capacity (especially the blue bars for adsorption). Depending on the application it looks like a steady loss of working capacity. Therefore it isn't perfectly stable under these conditions (but might be better than other MOFs). Consider rewording further. But otherwise this is fine for publication.

Revised. MIL-160 shows the retention of 99% total adsorption capacity after 16 cycles of sorption and the reviewer is correct that due to the accumulation of residual ammonia, a slight reduction of working capacity is observed (Figure 3a). A complete regeneration of MIL-160 with full retention of the porosity can be achieved (Figure 3b and 3c). Figure 3a has been updated with improved clarity and the comparison of stability towards NH₃ adsorption with other reported materials (J. Am. Chem. Soc. 2021, 143, 17, 6586–6592; J. Am. Chem. Soc. 2022, 144, 19, 8624–8632; J. Am. Chem. Soc. 2022, 144, 41, 18967–18975) add in the manuscript.

Reviewer #3 (Remarks to the Author):

In the revised manuscript, the authors addressed all points raised by the different referees. Moving some figures from the supplementary to the main text is clearly beneficial for the article.

There is just one point remaining that - after changing according to a referee's question - is still confusing. Bottom of page 8, the changed text to the 27Al spectra. First, I think the sentence structure is wrong, at least I don't understand the grammar. Second, the content is confusing me. If there are distorted octahedral moieties, then there is no long range order which is shown by the PXRD. So NMR and PXRD don't coincide. Was the PXRD done from the same material that was left in the MAS rotor for one week or a similar sample stored NH₃ for a week? In the latter case it might be that different samples are compared with NMR and PXRD. Otherwise I would be very surprised. The NMR spectrum shows that there is practically no 'crystalline' signal, so the overall content has to be very small. The PXRD shows just slight broadening of the peaks. Is there some indication that the 'intensity' is much lower in the PXRD compared to the other samples so that PXRD just 'sees' the tiny amount of crystalline parts remaining? From my experience, the former case is quite often occurring... The authors should comment on this, even if they write that they cannot explain the confusing results between NMR and PXRD. With that, the manuscript should be ready to go.

Revised. This is a very good point. Upon dosing NH₃ in MIL-160 for one week under ambient conditions (it is not the exactly the same sample from ssNMR measurement but from the same batch of synthesis), broadened Bragg peaks at low angle and decreased intensity are both observed from the PXRD patterns, indicating the partial loss of crystallinity (Figure 3b). This is in line with the distorted octahedral moieties observed by ssNMR. The discussion has been revised. We apologise for the confusion caused in previous version.